# Nimodipine Treatment Protects Auditory Hair Cells from Cisplatin-Induced Cell Death Accompanied by Upregulation of LMO4

**DOI:** 10.3390/ijms23105780

**Published:** 2022-05-21

**Authors:** Saskia Fritzsche, Christian Strauss, Christian Scheller, Sandra Leisz

**Affiliations:** Department of Neurosurgery, Medical Faculty, Martin Luther University Halle-Wittenberg, Ernst-Grube-Str. 40, 06120 Halle, Germany; saskia.fritzsche@uk-halle.de (S.F.); christian.strauss@uk-halle.de (C.S.); scheller1310@gmx.de (C.S.)

**Keywords:** nimodipine, cisplatin, ototoxicity, otoprotection, hearing, auditory hair cells, LMO4, Stat3

## Abstract

Ototoxicity is one of the main dose-limiting side effects of cisplatin chemotherapy and impairs the quality of life of tumor patients dramatically. Since there is currently no established standard therapy targeting hearing loss in cisplatin treatment, the aim of this study was to investigate the effect of nimodipine and its role in cell survival in cisplatin-associated hearing cell damage. To determine the cytotoxic effect, the cell death rate was measured using undifferentiated and differentiated UB/OC−1 and UB/OC−2 cells, after nimodipine pre-treatment and stress induction by cisplatin. Furthermore, immunoblot analysis and intracellular calcium measurement were performed to investigate anti-apoptotic signaling, which was associated with a reduced cytotoxic effect after nimodipine pre-treatment. Cisplatin’s cytotoxic effect was significantly attenuated by nimodipine up to 61%. In addition, nimodipine pre-treatment counteracted the reduction in LIM Domain Only 4 (LMO4) by cisplatin, which was associated with increased activation of Ak strain transforming/protein kinase B (Akt), cAMP response element-binding protein (CREB), and signal transducers and activators of transcription 3 (Stat3). Thus, nimodipine presents a potentially well-tolerated substance against the ototoxicity of cisplatin, which could result in a significant improvement in patients’ quality of life.

## 1. Introduction

Nimodipine belongs to the group of 1,4-dihydropyridines and exerts its effects by binding to the alpha1 subunit of L-type calcium channels, thereby decreasing calcium influx via negative allosteric inhibition [1,2]. Although nimodipine is one of the first calcium channel antagonists to be developed, it has been the focus of medical and scientific attention again [1,3,4,5,6]. Due to its good cerebrospinal fluid penetrability [2], which distinguishes it from other calcium antagonists of its substance class, nimodipine acts in the central nervous system. Because of this, nimodipine is routinely used in the clinic for the prophylaxis of cerebral vasospasm after subarachnoid hemorrhage by relaxing the smooth muscle of cerebral blood vessels, causing vasodilatation [1,3]. Nimodipine has also been administered in the area of ischemic stroke, traumatic brain injury, and migraine to investigate whether treatment leads to a positive outcome, but with heterogeneous results [1]. However, nimodipine has already shown a neuroprotective tendency and a beneficial effect on hearing preservation after vestibular schwannoma surgery [4,5,7,8].

Mentionable previous studies of our group showed a protective effect of nimodipine [9,10,11] pre-treatment on Schwann cells and neuronal cells, which was associated with increased phosphorylation of Akt and CREB and decreased activation of effector caspases [6]. Activation of Akt [12] and CREB [13] signaling is known to be involved in neuroprotection, leading to inhibition of caspase 3 activation and consequent prevention of apoptosis [14,15,16].

Cisplatin, a representative of platinum-based chemotherapeutics is still widely used in the antitumor therapy of many solid tumors, including testicular, ovarian, bladder, non-small cell lung carcinoma, and head and neck tumors [17,18,19]. Due to its molecular structure, the platinum derivative cisplatin is able to cross the plasma membrane and interacts with a variety of proteins but is also passively taken up into the cell via association with membrane transporters, for example, mammalian copper transporter 1 (CTR1) and organic cation transporter 2 (OCT2) [17,20,21,22]. Its cytotoxic effect consists of the formation of DNA-platinum adducts that prevent both DNA replication and RNA transcription. In addition, there is disruption of mitochondrial functions, activation of the immune system, and various other signaling pathways that also lead to cell death by apoptosis [17,20]. However, this cytotoxic effect is not limited to tumor cells and has toxic effects on healthy cells, which could lead to significant side effects. These include neurotoxicity, ototoxicity, and nephrotoxicity [17,23], which limit the dose and treatment time of cisplatin, due to their resulting reduction in patient quality of life [17]. Disorders such as cisplatin-induced peripheral neuropathy (CIPN) [17,24,25] and hearing impairment [18] require increased research into molecular mechanisms [20,22] and potential targets for intervention [19,24] to improve the therapeutic conditions for patients. Investigations of cisplatin-induced ototoxicity have shown that activation of the enzyme nicotinamide adenine dinucleotide phosphate (NAPDH)-oxidase 3 (NOX3), and thus the formation of reactive oxygen species (ROS) [23,26], contribute to toxicity. The resulting oxidative stress leads to the nitration of many proteins with LMO4 as its main target [23]. LMO4 is a transcriptional regulator [27] and, through the formation of transcription complexes, for example with CREB [28], it plays an important role in auditory hair cell survival [29] and inner ear development [27,30]. LMO4 knock-out leads to malformations of the organ of Corti, which is known as the ectopic organ of Corti (eOC) [30]. LMO4 function also has a protective effect through regulation of intracellular calcium concentration via expression of ryanodine receptor type 2, as the maintenance of homeostasis is essential for membrane potential, cell signaling pathways, and many others [28,31]. Further, LMO4 is known to be involved in anti-apoptotic signaling pathways, and the protein stabilizes glycoprotein-130, a subunit of the interleukin-6 (IL-6) receptor, which is thereby able to activate the Janus kinases (Jak)/Stat signaling pathway [18,32,33].

As it is one of the main proteins nitrated and diminished expressed by oxidative stress induced by cisplatin, LMO4 has been linked to the main side effects [18,23]. Therefore, the aim of this study was to investigate the effect of nimodipine in association with LMO4 expression and its central role in cell survival in cisplatin-associated auditory cell death. As there is currently no established therapy for hearing loss during cisplatin therapy, nimodipine could represent a potential medication to protect hair cells from apoptosis.

## 2. Results

### 2.1. Increase in Specific Hair Cell Markers after Differentiation of UB/OC−1 and UB/OC−2

The murine cell lines UB/OC−1 (Figure 1a,c) and UB/OC−2 (Figure 1b,d) were transferred from the undifferentiated (Figure 1a,b) to the differentiated (Figure 1c,d) state by culturing at 39 °C as described in methods and materials. In cell culture, the differentiation of both cell lines was visible through different morphology as described before [34]. 

Specific hair cell markers were detected to verify differentiation by performing qPCR. An increase in the *brain-specific homeobox/POU domain protein 3.1* (*Brn3.1*), *Myosin 6* (*Myo6*), *Myosin 7a* (*Myo7a*), and also *α9 acetylcholine receptor* (*α**9AChR*) gene expression was shown in both types of hair cells compared to its level in the undifferentiated state (set to 1.0, Table 1). 

### 2.2. Nimodipine Decreases Cisplatin-Induced Cytotoxicity in Undifferentiated Hair Cells

Activity of lactate dehydrogenase (LDH) in cell culture supernatant was used for cytotoxicity measurement induced by 20 µM, 50 µM and 100 µM cisplatin. The results showed a nimodipine-dependent reduction in cell death rate for both UB/OC−1 and UB/OC−2 cells (Figure 2a,b). Without stress induction by cisplatin, there was no cytotoxic effect induced by cisplatin’s solvent control (0.9% sodium chloride, NaCl, Figure 2a,b). Additionally, no reduction in cell death rate was measured between nimodipine’s solvent control absolute ethanol (EtOH) and the untreated control regardless of stress condition. In comparison to control cells (EtOH), nimodipine-treated UB/OC−1 cells showed a tendential but non-significant reduction at 20 µM and 50 µM cisplatin from 17.5% ± 5.9% to 12.1% ± 6.4% (10 µM nimodipine, not significant (n.s.)) and 9.3% ± 4.6% (20 µM nimodipine, n.s.) and from 22.5% ± 2.8% to 14.3% ± 4.0% (10 µM nimodipine, n.s.) and 12.6% ± 5.4% (20 µM nimodipine, n.s.). In addition, a decrease in cell death from 27.1% ± 4.6% to 13.0% ± 0.5% (10 µM nimodipine, *p* < 0.05) and 10.5% ± 0.6% (20 µM nimodipine, *p* < 0.05) at 100 µM cisplatin for UB/OC−1 cells (*p* < 0.05, Figure 2a) was measured.

The analysis of UB/OC−2 cells after nimodipine pre-treatment and stress induction demonstrated a significant reduction in cytotoxicity at 20 µM cisplatin from 26.9% ± 2.5% to 16.8% ± 1.6% (10 µM nimodipine, *p* < 0.05, Figure 2b) and 12.0% ± 1.4% (20 µM nimodipine, *p* < 0.05) and at 50 µM cisplatin from 23.1% ± 2.3% to 13.5% ± 1.0% (10 µM nimodipine, *p* < 0.05) and 12.6% ± 2.0% (20 µM nimodipine, *p* < 0.05). At 100 µM cisplatin cell death decreased from 28.3% ± 2.5% to 17.0% ± 0.9% (10 µM nimodipine, *p* < 0.05) and 11.9% ± 1.2% (20 µM nimodipine, *p* < 0.05) in comparison to cells treated with EtOH. Further multiple statistical comparisons are listed in Appendix A.

### 2.3. Nimodipine Decreases Cisplatin-Induced Cytotoxicity in Differentiated Hair Cells

After differentiation, a similar reduction in cytotoxicity was determined for both cell lines dependent on nimodipine pre-treatment (Figure 3a,b). No cytotoxic effect was measured in cells treated with 0.9% NaCl, whereas the untreated cells and the solvent EtOH treated cells showed nearly no LDH activity in the cell culture supernatant (Figure 3a). However, UB/OC−1 cells treated with 20 µM cisplatin and nimodipine showed a decrease in cell death from 36.3% ± 4.3% to 20.4% ± 5.6% (10 µM nimodipine, *p* < 0.05) and 14.1% ± 5.6% (20 µM nimodipine, *p* < 0.05). For cells treated with 50 µM cisplatin and nimodipine a non-significant reduction in cell death from 46.3% ± 13.2% to 32.7% ± 12.7% (10 µM nimodipine, n.s.) and 23.5% ± 7.1% (20 µM nimodipine, n.s.) (Figure 3a) was determined. A reduction at 100 µM cisplatin from 74.3% ± 9.3% to 48.4% ± 13.9% (10 µM nimodipine, n.s.) and 37.7% ± 4.9% (20 µM nimodipine, *p* < 0.05) for UB/OC−1 (Figure 3a) after nimodipine pre-treatment was measured. The differentiated nimodipine pre-treated UB/OC−2 cells showed lower cytotoxicity under stress induced by 20 µM, 50 µM, and 100 µM cisplatin (Figure 3b) than the untreated cells.

No significant reduction was detected between EtOH treated cells and control cells. Cells without stress (0.9% NaCl) showed an increase in LDH level at both 10 µM and 20 µM nimodipine to 27.0% ± 10.7% (n.s., Figure 3b) and 28.7% ± 6.6% (*p* < 0.05). Through nimodipine treatment, a decrease in cell death from 46.4% ± 7.8% to 27.2% ± 2.9% (10 µM nimodipine, *p* < 0.05) and 22.6% ± 3.1% (20 µM nimodipine, *p* < 0.05) and from 69.8% ± 2.3% to 51.6% ± 14.7% (10 µM nimodipine, n.s.) and 50.0% ± 6.6% (20 µM nimodipine, *p* < 0.05) cells treated with 20 µM and 100 µM cisplatin was shown. However, only a lower reduction was measured from 18.4% ± 4.3% to 15.0% ± 6.7% (10 µM nimodipine, n.s.) and 14.2% ± 6.1% (20 µM nimodipine, n.s.) after treatment with 50 µM cisplatin (Figure 3b). 

### 2.4. Nimodipine Counteracts the Downregulation of LMO4 by Cisplatin in Undifferentiated and Differentiated State of Hair Cells

Immunoblot analysis showed a strong reduction in LMO4 after cisplatin treatment (Figure 4). Under 20 µM cisplatin, an increase in the amount of LMO4 with 10 µM nimodipine and 20 µM nimodipine (Figure 4a,b) was detected, which is evident in both undifferentiated UB/OC−1 and UB/OC−2 cells. Additionally, differentiated UB/OC−1 and UB/OC−2 cells showed a strong decrease in LMO4 concentration measured after treatment with 20 µM cisplatin (Figure 4c, d). LMO4 remained constant without stress regardless of nimodipine treatment. Glyceraldehyde-3-phosphatedehydrogenase (GAPDH) served as a loading control.

### 2.5. Activation of Anti-Apoptotic Pathways by Nimodipine under Chemotherapy with Cisplatin in Undifferentiated and Differentiated State of Hair Cells

Undifferentiated UB/OC−1 and UB/OC−2 cells treated with 10 µM and 20 µM nimodipine showed increased phosphorylation of Akt at serine residue 473 and phosphorylation of CREB at serine residue 133, without and during stress conditions (Figure 5a,b) [6]. Increased activation of Stat3 through phosphorylation at tyrosine residue 705 by nimodipine pre-treatment is only evident after induction of stress with 20 µM cisplatin, while there is no altered phosphorylation detected without stress (Figure 5a,b). The total amount of Akt and CREB were not affected by either nimodipine or cisplatin treatment, whereas Stat3 protein level was reduced through stress induction with 20 µM cisplatin and increased through nimodipine pre-treatment without and with cell stress.

In the differentiated state, the immunoblots detected an increase in phosphorylation of Akt at serine residue 473 and CREB at serine residue 133 under 20 µM cisplatin, which increased after 10 µM and 20 µM nimodipine pre-treatment for both UB/OC−1 (Figure 5c) and UB/OC−2 (Figure 5d). The total amount of Akt and CREB were not affected by either nimodipine or cisplatin treatment. Stress induction by 20 µM cisplatin detected again a reduction in Stat3 activation, with 10 µM and 20 µM nimodipine pre-treatment leading to an increase in phosphorylation. The total amount of Stat3 was decreased by stress induction and showed an increase by nimodipine treatment without and during stress conditions in both UB/OC−1 and UB/OC−2 (Figure 5c,d).

### 2.6. Influence of Nimodipine on the Intracellular Calcium Concentration of Auditory Hair Cells under Cisplatin Treatment

The effect of different concentrations of nimodipine and cisplatin on the intracellular calcium concentration was investigated using the fluorescent dye Cal-520, which detects free calcium. The solvent control of nimodipine (EtOH) was set to 100% and compared with nimodipine-treated cells depending on the stress induction with cisplatin (Figure 6). In the investigation of undifferentiated UB/OC-1, a significant reduction in intracellular calcium concentration to 39.1% ± 17.0% (10 µM nimodipine, *p* < 0.05) and 22.6% ± 11.2% (20 µM nimodipine, *p* < 0.05) was observed at 0.9% NaCl, just as cells treated with 20 µM cisplatin to 57.5% ± 16.2% (10 µM nimodipine, *p* < 0.05, Figure 6a) and 36.2% ± 14.0% (20 µM nimodipine, *p* < 0.05) compared with the solvent control. Treatment with 50 µM cisplatin resulted in a decrease to 83.9% ± 7.2% (10 µM nimodipine, p < 0.05) and 63.5% ± 15.1% (20 µM nimodipine, *p* < 0.05). In comparison to EtOH-treated cells, nimodipine-treated UB/OC−2 cells showed a decrease in the amount of intracellular calcium at 0.9% NaCl to 85.9% ± 9.8% (10 µM nimodipine, n.s.) and 69.9% ± 2.9% (20 µM nimodipine, *p* < 0.05), while nearly no reduction to 99.3% ± 8.8% (10 µM nimodipine, n.s.) and 96.2% ± 4.6% (20 µM nimodipine, n.s.) was detected at 20 µM cisplatin (Figure 6b). The analysis after treatment with 50 µM cisplatin presented an increase in concentration to 116.2% ± 10.4% (10 µM nimodipine, n.s.) and 118.9% ± 8.7% (20 µM nimodipine, *p* < 0.05). In both data analyses, it can be seen that the reduction in calcium by nimodipine after cisplatin treatment is less effective and even leads to an increase when looking at the data of UB/OC−2. Further multiple statistical comparisons are listed in Appendix A.

## 3. Discussion

Cisplatin is widely used in antitumor therapy of solid tumors such as testicular, ovarian, bladder, non-small cell lung carcinoma, and head and neck tumors [17,18,19]. The main side effects of cisplatin treatment include neuropathy and hearing loss [17,20]. As a calcium channel antagonist with lipophilic properties [2], nimodipine has the possibility of acting centrally by crossing the blood–brain barrier and has already shown evidence of a neuroprotective effect in clinical trials [6,9,10,35] including a positive benefit on the preservation of hearing after surgery [6,8]. In parallel, in vitro studies on neuronal and Schwann cells showed a lower cytotoxic effect after nimodipine pre-treatment under different stress conditions [6,36]. The use of nimodipine has been established over the last 40 years, and its good tolerability, already proven in vivo, suggests that in addition to the standard treatment of vasospasm following subarachnoid hemorrhage, the use of nimodipine in clinical practice can be further extended [1,2,3]. In this study, the effect of nimodipine on cisplatin-induced apoptosis of inner ear hair cells was investigated for the first time. Increasingly with the concentration of nimodipine, we also confirmed a significant reduction in cytotoxicity in both cochlear pre-cursor cell lines UB/OC−1 and UB/OC−2 during cisplatin stress. While previous studies observed a dose-independent effect [6,36,37], this study showed an increase in the protective effect with increasing nimodipine concentration. Further studies should shed more light on whether higher concentrations of nimodipine lead to a further reduction in cytotoxicity. Here, a stronger effect of the calcium channel antagonist under cisplatin was observed in the undifferentiated UB/OC−2 cells (up to 58.0% compared to 39.1%) while the effect was similar in differentiated and undifferentiated UB/OC−1 cells (up to 61.2% compared to 61.3%). Furthermore, the results showed an increase in sensitivity of the cells after differentiation, as evidenced by a marked increase in cell death under cisplatin after differentiation. At this stage of development, hair cell lines are most comparable to conditions in vivo [34], clearly demonstrating the impact of cisplatin on the inner ear. Whether a hyperthermal effect contributed to the increased apoptosis of the cells under treatment with cisplatin should be further investigated with additional thermal stress. It was striking that the cytotoxic effect of 20 µM cisplatin was more pronounced in the UB/OC−2 cells at both levels of differentiation compared to stress with 50 µM cisplatin. Further investigations should shed more light on whether any protective mechanisms are only activated after a certain degree of stress or whether other intracellular signaling pathways play a role in this. However, the results mainly demonstrated incidents in cell culture models of isolated hair cells independent of the systemic influence of an organism. In the long term, further studies in murine animal models could provide information on whether this effect can be confirmed in vivo. In addition, the impact of nimodipine treatment in combination with cisplatin should be tested on malignant cells. Preliminary studies on several tumor cell lines indicate that there is no protective effect of nimodipine on malignant cells (data not shown).

The formation of reactive oxygen and nitrogen species leads to the downregulation of the transcription regulator LMO4, during chemotherapy with cisplatin (Figure 7a) [18,23,26,32]. This is associated with a negative influence on cell survival of cisplatin treatment [26]. As one of the main side effects of chemotherapy with cisplatin, ototoxicity is also related to the downregulation of LMO4 [18,26,32]. This protein is found in three regions of the cochlea: the spiral ganglion, the organ of Corti, and the stria vascularis [32]. It also plays an important role in the development of the inner ear and knock-out leads to a malformation of the organ of Corti in mice [30]. Since a physiological protein level of LMO4 causes intact cell function in terms of anti-apoptotic effect [18], our study focused on the influence of nimodipine under cisplatin on LMO4. Pre-treatment of immortalized hair cells with 10 µM and 20 µM nimodipine counteracts the downregulation of LMO4 and thus leads to an increase under cisplatin stress (Figure 4 and Figure 7b). This positive regulation of LMO4 through nimodipine treatment has been shown also for the first time and could represent a key factor in nimodipine’s otoprotective mechanism of action. Further studies should shed light on whether there is a protective effect of nimodipine even after the knock-out of LMO4 and whether this also counteracts the negative effects of cisplatin to the same extent in vivo.

As in previous studies on neuronal and Schwann cells, nimodipine pre-treatment led to increased activation of Akt at the serine residue 473 as well as increased activation of the transcription factor CREB at serine residue 133 by phosphorylation [6]. Both cell signaling proteins are known to interact with LMO4 [38] and to promote an anti-apoptotic effect (Figure 7) [12,13,39]. Increased expression of LMO4 leads to higher activation of Akt, which exerts a neuroprotective effect via its various anti-apoptotic pathways [12,38,39]. A higher protein level of LMO4 also increases the phosphorylation of transcription factor CREB [40], resulting in neuronal cell survival during the interaction with Akt [13,14] as well as with LMO4 via the formation of a transcription complex [28]. CREB exerts its neuroprotective effect via starting gene expression with anti-apoptotic activity, such as B-cell lymphoma 2 (Bcl2) acting via regulation of intrinsic apoptosis induction and promoting DNA damage repair [40]. Therefore, these results clearly demonstrate the positive effect of nimodipine on cell survival of hair cells, which is accompanied by upregulation of LMO4 and activation of Akt and CREB (Figure 7b). Further functional studies should elucidate whether LMO4 plays a modulatory role in the activation of CREB and Akt or the mechanism of action of nimodipine.

Furthermore, higher expression of LMO4 can lead via the stabilization of glycoprotein 130 to increased activation of its downstream target Stat3 via the Jak/Stat pathway. This is known to cause increased expression of anti-apoptotic genes and thus downregulation is associated with apoptosis and thus ototoxicity [18,33]. Rosati et al. 2019 already showed a strong decrease in Stat3 and less activation by cisplatin. In our results, nimodipine was shown not only to increase LMO4 but also to increase the activation of Stat3 at tyrosine residue 705 under stress after it was strongly reduced by the influence of cisplatin (Figure 7). In addition, an increased nimodipine-dependent total protein level of Stat3 was detected independent of the stress condition.

The maintenance of intracellular calcium homeostasis plays an essential role in a lot of important cell functions and cell survival and is regulated by a complex system [41,42]. Nimodipine regulates the calcium influx as an L-type calcium channel inhibitor [1,2]. Since it has already been shown that the calcium concentration increases under cisplatin [18,20], it was hypothesized that the neuroprotective effect of nimodipine is achieved by preventing calcium overload of the cell and stabilizing calcium homeostasis. In contrast, previous studies suggested a toxic effect of incoming calcium triggered via N-methyl-d-aspartate (NMDA) receptor, an ionotropic glutamate receptor, rather than L-type calcium channels [42,43]. Our study showed an opposite effect in immortalized hair cells, where divergent results between the different cell lines indicate a more receptor-independent effect. While UB/OC−1 cells showed a decrease in intracellular calcium concentration by nimodipine under cisplatin stress, no significant change in intracellular calcium was observed in UB/OC−2 cells, which was also confirmed for the differentiated cells (data not shown). The assumption that reduced intracellular calcium concentration is caused by a reduced cell quantity must be rejected since otherwise there would be no increase in undifferentiated UB/OC−2 or in differentiated cells under stress, while the cell number has been reduced by apoptosis. Since it is also known that LMO4, in addition to its own regulation by calcium, also has an influence on calcium concentration via the expression of ryanodine receptors [28,31], further investigations should aim to determine whether a protective effect occurs in a calcium-dependent manner and in combination with nimodipine pre-treatment.

In summary, it can be concluded that nimodipine reduces the cytotoxic effect induced by cisplatin accompanied by the upregulation of LMO4 and the associated activation of anti-apoptotic pathways in vitro (Figure 7).

Clinical trials focusing on the treatment of ototoxic effect triggered by cisplatin have already been carried out numerous times in recent years and, since no standard therapy has yet been approved by the Federal Drug Administration, many studies with a potentially good outcome are currently underway [19]. Otoprotection by SENS-401 (R-azasetron besylate) was observed in both in vitro and in vivo models with no effect on the chemotherapeutic potential of cisplatin [44,45]. Amifostine initially showed otoprotective potential in average-risk medulloblastoma patients [46,47], but this was found to be insufficient upon further investigation [48]. Similarly, the promising inhibition of OCT2 [21] by the proton pump inhibitor pantoprazole only leads to an insufficient reduction [49]. The intratympanic injection of dexamethasone, in contrast to systemic administration [50], showed a protective effect in cisplatin-induced ototoxicity [51]. Further studies on new non-invasive forms of administration observed the same effect [52,53]. For nimodipine, too, with the same otoprotective efficacy in vivo, studies should be conducted on the best possible form of administration. Substances such as the antioxidant N-acetylcysteine also counteract hearing loss during chemotherapy with cisplatin [54,55] and, like ginkgo [56] and other substances, are the focus of current clinical studies. In particular, sodium thiosulphate stood out as a potential agent that is both well tolerated [57] and, in a phase III clinical trial, showed a significant reduction in the incidence of hearing loss during chemotherapy with cisplatin in children with standard-risk hepatoblastoma without compromising the chemotherapeutic potential [58]. Despite the large number of substances that have already been investigated, there is still no established standard therapy against cisplatin-based ototoxicity. The results in neuroprotection of nimodipine, as well as its good tolerability, identifies it as a potential new medication.

By reducing the side effects of cisplatin through nimodipine pre-treatment, a significant improvement in patients’ quality of life and better utilization of the chemotherapeutic effect could be achieved, if this protective effect can also be demonstrated in vivo. However, the most effective therapy against the side effects of cisplatin is not only the molecular inhibition of intracellular apoptosis mechanisms but also the optimization of the tumor-centered effect and blockade of the formation and impact of the toxic degradation products of cisplatin [59]. In addition to the reduced compliance of patients with the increase in side effects under chemotherapy, these also lead to a deterioration in mental health. Likewise, the removal or reduction in the dose limitation leads to greater potential in the total utilization of the chemotherapeutic effect of platinum derivatives. Nimodipine could thus play a potential role in the improved use of chemotherapeutic agents such as cisplatin by improving the quality of life of patients under and after oncological disease.

## 4. Materials and Methods

### 4.1. Cell Lines

The murine cell lines UB/OC−1 (CVCL_9636, organ of Corti 1) and UB/OC−2 (CVCL_D790, organ of Corti 2) were obtained from Ximbo (London, UK) and have been established by Matthew Holly (University of Bristol, UK). Both were cultured in Eagle’s Minimum Essential Medium (EMEM, LONZA, Basel, Switzerland) supplemented with 10% fetal bovine serum (FBS, Gibco, Thermo Fisher Scientific, Waltham, MA, USA), 100 U/mL penicillin, and 100 mg/mL streptomycin (Gibco, Thermo Fisher Scientific, Waltham, MA, USA), 2 mM GlutaMAX^TM^ (Gibco, Thermo Fisher Scientific, Waltham, MA, USA) and 50 Units/mL Interferon gamma (γ -IFN, ImmunoTools, Friesoythe, Germany) in undifferentiated state at 5% CO_2_ and 33 °C. For differentiation, cells were cultured as previously described [34] with supplemented EMEM except γ-IFN at 5% CO_2_ and 39 °C followed by verifying through gene expression analysis of the hair cell markers *Brn3.1*, *Myo6*, *Myo7a* and *α9AChR* after 14 days via quantitative polymerase chain reaction (qPCR). All experiments were performed in both differentiated and undifferentiated states (Figure 8, stages of development). At least three independent biological replicates were carried out for each experiment.

### 4.2. Cell Treatment

Cells were seeded and pre-treated with 10 µM and 20 µM nimodipine (Tokyo Chemical Industry, Zwijndrecht, Belgium) diluted in EtOH (1:1000, Carl Roth, Karlsruhe, Germany). Cells treated with equal amount of EtOH (0.1%, *v/v*) served as control (Figure 9a). After 24 h, cytotoxicity was induced with 20 µM, 50 µM, and 100 µM cisplatin (Sigma-Aldrich, St. Louis, MO, USA) diluted in 0.9% NaCl (B. BRAUN, Melsungen, Germany). Simultaneously, nimodipine was given in the same amount as the day before, respectively (Figure 9b). NaCl served as solvent control of cisplatin.

### 4.3. Cytotoxicity Measurement

The 5 × 10^4^ UB/OC−1 or UB/OC−2 were seeded in 24-well plates (Techno Plastic Products, TPP, Trasadingen, Switzerland) and cytotoxicity was measured by the activity of LDH as a marker for cell death using Cytotoxicity Detection Kit (Roche, Basel, Switzerland) according to manufacturer’s instructions. In brief, 100 µL cell culture supernatant in triplicates per sample and 100 µL reaction mix were incubated in the dark for 30 min. Absorbance was measured at 492 nm with Tecan Reader F2000 Pro (Tecan, Männedorf, Switzerland) at four definite points of the wells. Absorbance of cells lysed with 2% Triton X-100 (Carl Roth, Karlsruhe, Germany) served as positive control (100% cell death) while medium signal without cells served as background signal. The calculation of the cell death rate was performed as described before [6].

### 4.4. Western Blot

The 2 × 10^6^ cells were seeded in Petri dishes (TPP, Trasadingen, Switzerland) and treated as shown in Figure 9 schematically. At 24 h after cisplatin treatment, cells were washed two times with ice-cold Dulbecco’s Phosphate Buffered Saline (PBS, Thermo Fisher Scientific, Waltham, MA, USA) and harvested in PBS containing Halt^TM^ Protease and Phosphatase Inhibitor Single-Use Cocktail (1:100, Thermo Fisher Scientific, Waltham, MA, USA). The proteins were extracted with 1x LDS Sample Buffer (Invitrogen, Thermo Fisher Scientific, Waltham, MA, USA) and heated at 70 °C for 10 min, followed by protein concentration measurement performing bicinchoninic acid (BCA) assay using Pierce^TM^ BCA Protein Assay Kit (Thermo Fisher Scientific, Waltham, MA, USA) according to manufacturer’s instructions. After 5% *β*-mercaptoethanol (Carl Roth, Karlsruhe, Germany) was added, the samples were again heated at 70 °C for 10 min. To separate protein, SDS-PAGE was performed using NuPAGE^TM^ 4-12% Bis-Tris Gels (1.5 mm × 10 well) (Invitrogen, Thermo Fisher Scientific, Waltham, MA, USA) and 1× NuPAGE^®^ MES SDS Running Buffer (Novex, Thermo Fisher Scientific, Waltham, MA, USA) followed by blotting onto 0.2 µm or 0.45 µm nitrocellulose membranes (Amersham, GE, Healthcare, Freiburg, Germany) depending on molecular weight of the proteins to be detected (0.2 µm < 20 kDa). After blocking with 5% skim milk (Carl Roth, Karlsruhe, Germany) diluted in tris-buffered saline (TBS) with 0.1% Tween^®^ 20 (Sigma-Aldrich St. Louis, MO, USA, TBS-T), membranes were incubated overnight at 4 °C with primary antibodies (Table 2). Afterward, membranes were washed five times with TBS-T for 5 min, incubated with the secondary horseradish peroxidase (HRP)-linked antibody (Table 2) for at least 60 min, and washed again three times with TBS-T and two times with TBS for 5 min each. Membranes were developed using Pierce^TM^ ECL Western Blotting Substrate (Thermo Fisher Scientific, Waltham, MA, USA) and signals were detected with a CCD camera (ImageQuant LAS4000, GE, Healthcare, Freiburg, Germany).

### 4.5. Intracellular Calcium Measurement

The 1 × 10^4^ cells were seeded in black flat 96-well plates (Greiner Bio-One, Kremsmünster, Austria). For measurement of intracellular calcium concentration Cal-520 No Wash Calcium Assay (Abcam, Cambridge, UK) was used. Five µM of the fluorescent dye Cal-520 (Abcam, Cambridge, UK) were added to Hank’s Balanced salt solution (HBSS, Gibco, Thermo Fisher Scientific, Waltham, MA, USA) supplemented with 20 mM 4-(2-hydroxyethyl)-1-piperazineethanesulfonic acid (HEPES) buffer (LONZA, Basel, Switzerland), 0.02% Pluronic^TM^ F-127 (Invitrogen, Thermo Fisher Scientific, Waltham, MA, USA) and 1 mM Probenecid (Sigma-Aldrich St. Louis, MO, USA) from which 100 µL were given to each well. Afterward, incubating the plate for 90 min at 33 °C or 39 °C and adding 100 µL Probenecid dissolved in Minimum Essential Medium (MEM, Gibco, Thermo Fisher Scientific, Waltham, MA, USA) intracellular calcium concentration was measured at Ex/Em = 490/525 nm with Tecan infinite F2000 Pro (Tecan, Männedorf, Switzerland).

### 4.6. RNA Isolation and Real-Time Quantitative PCR

RNA was isolated with NucleoSpin^®^ RNA Plus (MACHERY-NAGEL, Düren, Germany) following manufacturer’s instructions. The RevertAid First Strand cDNA Synthesis Kit (Thermo Fisher Scientific, Waltham, MA, USA) was used to transcribe 2 µg of RNA into cDNA. 20 ng cDNA, 5 µM specific primer (Table 3), and 1× Platinum^®^ SYBR^®^ Green qPCR SuperMix-UDG (Invitrogen, Thermo Fisher Scientific, Waltham, MA, USA) in a total volume of 20 µL were prepared respectively. qPCR was performed with Rotor-Gene Q (Qiagen, Hilden, Germany).

### 4.7. Statistical Analysis

Statistical analysis of three independent replicates was performed with unpaired, two-sided student’s t-test and one-way ANOVA followed by Tukey’s post hoc test (SPSS version 28, IBM, Ehringen, Germany). Significance was accepted if *p* values were <0.05. Data were expressed as the mean ± S.D.

## 5. Conclusions

Nimodipine not only acts via vasodilation as a calcium channel antagonist but also shows a protective effect against cell death induced by cisplatin in auditory cells. This offers the possibility to extend the application of vasospasm prevention after subarachnoid hemorrhages to the treatment of prevention of cisplatin-mediated side effects or auditory cell damage associated with degenerative diseases.

The transcriptional regulator LMO4 and the accompanied increased activation of anti-apoptotic pathways via Akt, CREB, and Stat3 seemed to be associated with the otoprotective effect. Future studies should aim to prove the obtained results in vivo in order to enable future application in patients.

## Figures and Tables

**Figure 1 ijms-23-05780-f001:**
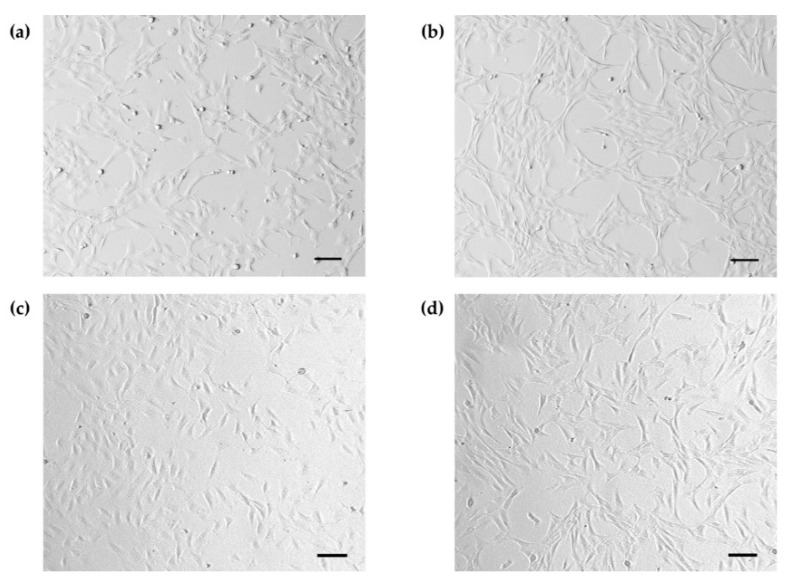
**Microscopic images of different auditory hair cell states.** Undifferentiated UB/OC−1 (**a**) and UB/OC−2 (**b**) cells were cultivated at 33 °C, whereas differentiating UB/OC−1 (**c**) and UB/OC−2 (**d**) cells were incubated at 39 °C for at least 14 days. The scale bar corresponds to 100 µm.

**Figure 2 ijms-23-05780-f002:**
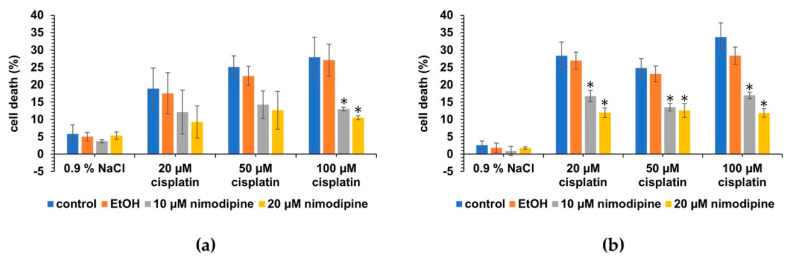
**Prevention of cisplatin**-**induced cytotoxicity by nimodipine in auditory hair cells.** The 5 × 10^4^ cells were seeded and treated with nimodipine, and cisplatin as described in the methods and materials section. To investigate the cell death rate, the LDH activity in the culture supernatant was measured 24 h after stress induction. A reduction in cell death induced by cisplatin treatment by pre-treatment with 10 µM and 20 µM nimodipine was visible in both UB/OC−1 (**a**) and UB/OC−2 (**b**) cells. Between cells treated with EtOH (0.1%, solvent) and untreated cells, there was no reduction in cytotoxicity. *p* values < 0.05 (* *p* < 0.05) compared to cells treated with EtOH death rates were accepted as significant. The mean values and standard deviations of three independent biological replicates are shown.

**Figure 3 ijms-23-05780-f003:**
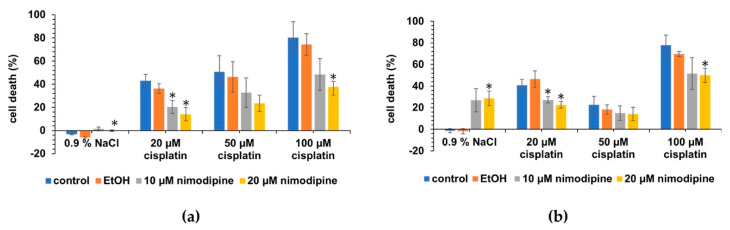
**Reduced cell death by nimodipine in differentiated hair cells.** UB/OC−1 (**a**) and UB/OC−2 (**b**) cells were pre-treated with 10 µM and 20 µM nimodipine and stressed with indicated concentrations of cisplatin the following day. A reduction in cytotoxicity in the nimodipine-treated cells was determined compared to EtOH-treated cells upon stress induction by cisplatin. *p* values < 0.05 (* *p* < 0.05) compared to the solvent control the cell death rates were accepted as significant.

**Figure 4 ijms-23-05780-f004:**
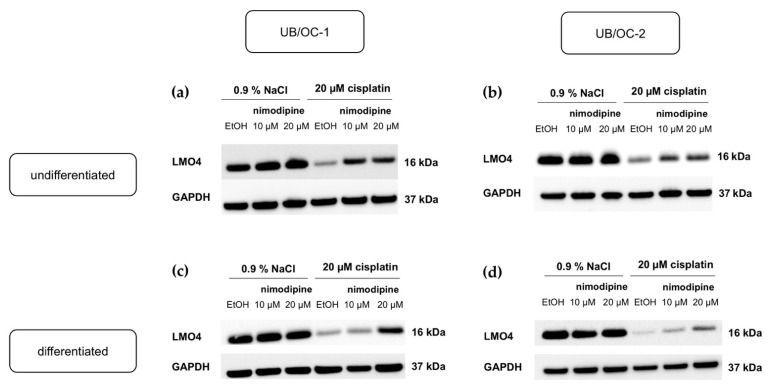
**Analysis of the LMO4 protein amount depending on nimodipine and cisplatin treatment.** A representative Western blot of three independent biological replicates of undifferentiated UB/OC−1 (**a**) and UB/OC−2 (**b**) cells as well as of differentiated UB/OC−1 (**c**) and UB/OC−2 (**d**) cells is shown. Fifty µg protein per sample was loaded and separated via sodium dodecyl sulfate-polyacrylamide gel electrophoresis (SDS-PAGE) and blotted onto 0.2 µm nitrocellulose membranes as described in materials and methods section. Detection was performed by using specific antibodies. While the amount of the protein LMO4 was strongly reduced by cisplatin, there was an increase by pre-treatment with 10 µM and 20 µM nimodipine during 20 µM cisplatin.

**Figure 5 ijms-23-05780-f005:**
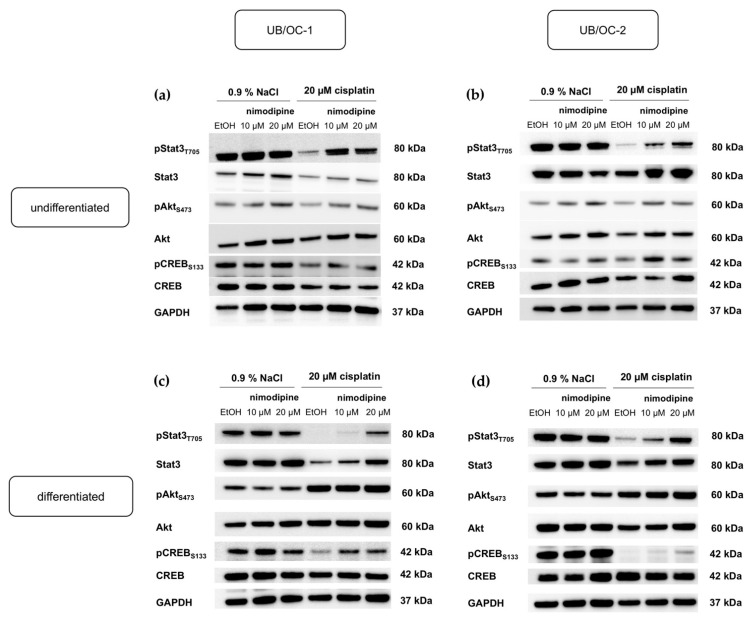
**Detection of anti-apoptotic pathways activated by nimodipine during cisplatin treatment for undifferentiated UB/OC−1 (a) and UB/OC−2 (b) cells as well as for differentiated UB/OC−1 (c) and UB/OC−2 (d) cells.** After transfer of the proteins separated by SDS-PAGE (30 µg) onto 0.45 µm nitrocellulose membranes, the phosphorylation and total protein amount of anti-apoptotic cell signaling components were determined by specific antibodies. As a loading control, GAPDH protein level was used. The Western blot shown is representative of the results from three independent biological replicates.

**Figure 6 ijms-23-05780-f006:**
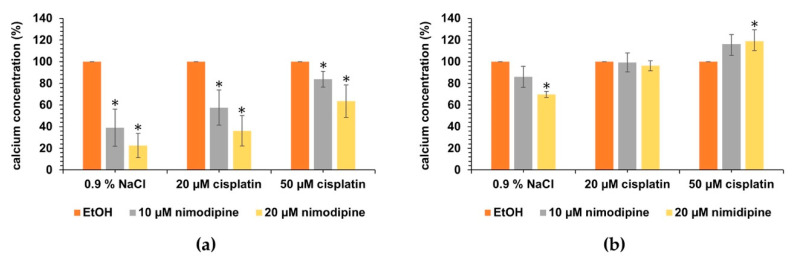
**Intracellular calcium amount under cisplatin stress of nimodipine pre-treated UB/OC−1** (**a**) **and UB/OC−2** (**b**) **cells.** The 1 × 10^4^ cells were seeded in black flat 96-well plates and treated as described in method section. Detection of intracellular calcium by the fluorescent dye Cal-520 showed a decrease by nimodipine pre-treatment in UB/OC−1 cells that was reduced with increasing cisplatin concentration. Examination of nimodipine-treated UB/OC−2 cells initially showed a decrease in the amount of calcium without stress induction (0.9% NaCl), which was minimally visible at 20 µM cisplatin and changed to an increase at 50 µM. The graphs show the results of three independent biological replicates. If the p values were <0.05 (* *p* < 0.05) compared to the solvent control cells (EtOH) were accepted as significant.

**Figure 7 ijms-23-05780-f007:**
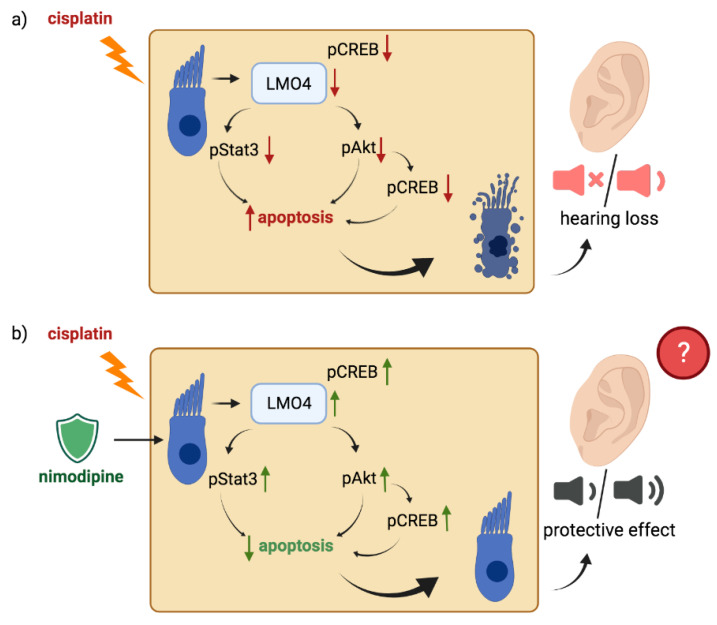
**Nimodipine pre-treatment could protect hair cells from cisplatin-induced apoptosis via upregulation of LMO4 and anti-apoptotic pathways.** Cisplatin leads to the downregulation of the transcription factor LMO4 via the production of intracellular reactive oxygen and nitrogen species. This causes an increase in apoptosis of auditory hair cells and hearing loss, one of the most common side effects of this chemotherapy, via downregulation of the anti-apoptotic cell signaling pathways (**a**). Nimodipine counteracts this process and reduces auditory hair cell death via the upregulation of LMO4 and the associated activation of Akt, CREB and Stat3 (**b**). The scheme was created with BioRender.com.

**Figure 8 ijms-23-05780-f008:**
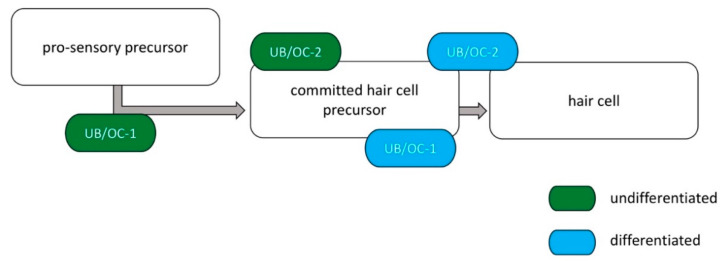
**Developmental stages of immortalized hair cells depending on the degree of differentiation.** In the undifferentiated state, the UB/OC−1 are at the stage of a pro-sensory progenitor, while the UB/OC−2 cells exhibit early characteristics of committed hair cell precursor cells. After differentiation, UB/OC−1 cells are in the process of differentiating into hair cells. The UB/OC−2 are at the stage of solid hair cells. The scheme is based on the model of Rivolta et al., 1998 [34].

**Figure 9 ijms-23-05780-f009:**
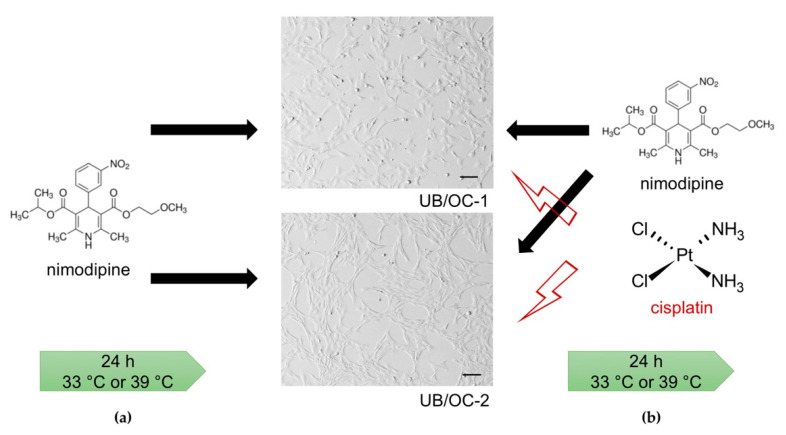
**Scheme of nimodipine and cisplatin treatment.** To investigate the protective effect of nimodipine on cisplatin-associated ototoxicity UB/OC−1 and UB/OC−2 were pre-treated with different concentrations of nimodipine for 24 h (**a**). Afterward, stress was induced by 20 µM, 50 µM, and 100 µM cisplatin while the same amount of nimodipine was added again. Experiments were performed 24 h after the cell stress was triggered (**b**). Microscopic images of UB/OC−1 and UB/OC−2 cells were created with Keyence BZ-X810 (Keyence, Neu-Isenburg, Germany).

**Table 1 ijms-23-05780-t001:** Upregulation of hair cell markers through cell differentiation.

	UB/OC−1	UB/OC−2
Gene Name	Factor of Upregulation after Differentation	
*Brn3.1*	2.5 ± 0.9	2.8 ± 1.9
*Myo6*	1.5 ± 0.9	2.7 ± 2.2
*Myo7a*	1.1 ± 0.2	9.9 ± 4.3
*α9AChR*	2.4 ± 0.9	5.3 ± 2.9

The data are the mean values and standard deviation (S.D.) from three independent biological replicates.

**Table 2 ijms-23-05780-t002:** Antibodies used for Western Blot.

Antibody	Isotype	Dilution	Dilution Buffer	Manufacturer
Phospho-Akt (Ser473) (D9E) XP^®^ #4060	Rabbit IgG	1:1000	5% BSA in TBS-T	Cell Signaling Technology (Danvers, MA, USA)
Akt (pan) (40D4) #2920	Mouse IgG1	1:2000	5% MP in TBS-T	
Phospho-CREB (Ser133) (87G3) #9198	Rabbit IgG	1:1000	5% MP in TBS-T	
CREB (48H2) #9197	Rabbit IgG	1:1000	5% BSA in TBS-T	
Phospho-Stat3 (Tyr705) (3E2) #9138	Mouse IgG1	1:1000	5% MP in TBS-T	
Stat3 (124H6) #9139	Mouse IgG2a	1:1000	5% MP in TBS-T	
LMO4 (D6V4Z) #81428	Rabbit IgG	1:1000	5% BSA in TBS-T	
GAPDH (14C10) #2118	Rabbit IgG	1:1000	5% BSA in TBS-T	
Anti-mouse IgG, HRP-linked Antibody	Horse	1:1000	2% MP in TBS-T	
Anti-rabbit IgG, HRP-linked Antibody	Goat	1:1000	2% MP in TBS-T	

BSA, bovine serum albumin (Carl Roth, Karlsruhe, Germany); GAPDH, glyceraldehyde-3-phosphatedehydrogenase; MP, skim milk powder; IgG, immunoglobulin G.

**Table 3 ijms-23-05780-t003:** Primer used for quantitative real-time PCR to verify differentiation.

Gene Name	Oligo Sequence 5′ to 3′ (Forward, Reverse)	Annealing Temperature	Reference Sequence	Species
*Brn3.1*	TTCAACGGCAGTGAGCGTAA, ACAGAACCAGACCCTCACCA	60 °C	NM_138945.2	*Mus musculus*
*Myo6*	CATGGCACTCCGAAGAGGT, GGGATTCTGCTGAGGTGAATTG	60 °C	NM_001039546.2	
*Myo7a*	ACTGCTCTGTGAGACATCGC, ACCAGGAAGGCCACAACAAA	60 °C	NM_001256081.1	
*α9AChR*	TGCACGCTATGAAGCACTGA, CGAATGCCTACCAACCCACT	60 °C	NM_001081104.1	
*GAPDH*	GCACAGTCAAGGCCGAGAAT, GCCTTCTCCATGGTGGTGAA	60 °C	NM_001289726	

## Data Availability

Not applicable.

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
