# Peer review of "Nimodipine Treatment Protects Auditory Hair Cells from Cisplatin-Induced Cell Death Accompanied by Upregulation of LMO4"

_ijms, 2022, doi:10.3390/ijms23105780_

Round 1
Reviewer 1 Report
Dear authors,
the manuscript you submitted is of interest for the use of cisplatin in chemotherapy therapy. Side effects such as ototoxicity appear to be reduced with the combination of nimodipine with cisplatin. This would increase the patient's well-being. The results are clear and comprehensive, the discussion is based on very recent and well written articles.
Reviewer 2 Report
- The study enumerating the protective effects of nimodipine against cisplatin induced toxicity has been reviewed.
- The work overall neat and free from major technical errors.
- The section 2.5 needs to be written more clearly with regards to the treatment inductions. I believe that this section is partly confusing for the readers.
- I can recommend the current article with minor revisions.
